# Outcomes of Conversion Surgery for Metastatic Gastric Cancer Compared with In-Front Surgery Plus Palliative Chemotherapy or In-Front Surgery Alone

**DOI:** 10.3390/jpm12040555

**Published:** 2022-04-01

**Authors:** Ruo-Yi Huang, Hao-Wei Kou, Puo-Hsien Le, Chia-Jung Kuo, Tsung-Hsing Chen, Shang-Yu Wang, Jen-Shi Chen, Ta-Sen Yeh, Jun-Te Hsu

**Affiliations:** 1Department of General Surgery, Chang Gung Memorial Hospital at Linkou, College of Medicine, Chang Gung University, Taoyuan 33305, Taiwan; mr1309@cgmh.org.tw (R.-Y.H.); b9602039@cgmh.org.tw (H.-W.K.); m7026@cgmh.org.tw (S.-Y.W.); tsy471027@cgmh.org.tw (T.-S.Y.); 2Department of Gastroenterology, Chang Gung Memorial Hospital at Linkou, College of Medicine, Chang Gung University, Taoyuan 33305, Taiwan; b9005031@cgmh.org.tw (P.-H.L.); m7011@cgmh.org.tw (C.-J.K.); q122583@cgmh.org.tw (T.-H.C.); 3Department of Hematology-Oncology, Chang Gung Memorial Hospital at Linkou, College of Medicine, Chang Gung University, Taoyuan 33305, Taiwan; js1101@cgmh.org.tw

**Keywords:** conversion surgery, metastatic gastric cancer, chemotherapy, outcomes

## Abstract

The survival benefits of conversion surgery in patients with metastatic gastric cancer (mGC) remain unclear. Thus, this study aimed to determine the outcomes of conversion surgery compared to in-front surgery plus palliative chemotherapy (PCT) or in-front surgery alone for mGC. We recruited 182 consecutive patients with mGC who underwent gastrectomy, including conversion surgery, in-front surgery plus PCT, and in-front surgery alone at Linkou Chang Gung Memorial Hospital from 2011 to 2019. The tumor was staged according to the 8th edition of the American Joint Committee on Cancer. Patient demographics and clinicopathological factors were assessed. Overall survival (OS) was evaluated using the Kaplan–Meier curve and compared among groups. Conversion surgery showed a significantly longer median OS than in-front surgery plus PCT or in-front surgery alone (23.4 vs. 13.7 vs. 5.6 months; log rank *p* < 0.0001). The median OS of patients with downstaging (pathological stage I–III) was longer than that of patients without downstaging (stage IV) (30.9 vs. 18.0 months; *p* = 0.016). Our study shows that conversion surgery is associated with survival benefits compared to in-front surgery plus PCT or in-front surgery alone in patients with mGC. Patients who underwent conversion surgery with downstaging had a better prognosis than those without downstaging.

## 1. Introduction

Gastric adenocarcinoma (GC) is the fifth most common malignancy worldwide and the third leading cause of cancer-related deaths, with a high incidence in South and East Asia [1]. Most patients are diagnosed at an advanced stage, with a high frequency of local invasion or metastasis, because of its asymptomatic nature in the early stages [2]. The first-line treatment of stage IV or metastatic GC (mGC) includes chemotherapy alone, immunotherapy alone, or a combination of chemotherapy and/or immunotherapy plus targeted therapy based on tumor characteristics and the patient’s general condition, with a median overall survival (OS) of 8–16 months [3,4,5,6,7,8]. The role of surgery for mGC is mainly to manage tumor-related complications, such as obstruction, bleeding, or perforation. Studies have indicated that palliative gastrectomy can improve the patient’s quality of life including normal activity, diet, and fewer gastrointestinal symptoms (such as nausea/vomiting or bleeding) [9].

The survival benefit of in-front palliative gastrectomy remains controversial [10,11,12]. Our previous retrospective study recruiting patients with mGC between 2000 and 2010 showed that palliative resection followed by palliative chemotherapy (PCT) was associated with a longer survival time compared to surgery alone, chemotherapy alone, or supportive care [10]. A retrospective study enrolling 5599 mGC cases from 2008 to 2015 and using the Taiwan Cancer Registry database also indicated that patients that received surgery plus chemotherapy had the longest survival compared to those who received other treatments [12]. However, a phase III study (REGATTA trial) indicated that palliative gastrectomy (D1 lymphadenectomy) followed by chemotherapy did not prolong survival compared to chemotherapy alone for mGC patients with a single risk factor for non-curativeness (liver, peritoneal surface, or para-aortic lymph node metastasis) [11]. Therefore, chemotherapy remains the mainstream treatment for mGC. However, in patients who respond to PCT, prolonged chemotherapy may induce acquired chemo-resistance or the cumulative side effects of PCT could eventually impact the efficacy of chemotherapy [13,14]. In this regard, a new therapeutic concept has been proposed, that surgery might be offered during the treatment as a part of a multimodal treatment strategy for select mGC patients, which is defined as “conversion surgery” [15,16,17,18,19]. Conversion surgery has been described as a potentially curative resection following induction chemotherapy for an initially unresectable or borderline resectable (for technical and/or oncological reasons) tumor [20,21,22].

Given the lack of an established global standard of care and the limited survival benefit obtained from currently recommended therapies for mGC, there is an urgent need for other treatment strategies. Although mounting evidence has shown improved survival with conversion surgery in selected patients [15,16,17,18,19], there is currently no global consensus recommending this approach for mGC, and a well-designed randomized trial relevant to this topic is not available at present. In addition, details regarding patients who may benefit more from this therapeutic approach remain unclear [23]. Thus, the current study aimed to explore the impact of conversion surgery on OS compared with in-front surgery plus PCT or in-front surgery alone at a tertiary medical center.

## 2. Methods

### 2.1. Patients

We recruited consecutive patients with clinically diagnosed mGC (stage IVB) who underwent gastrectomy at Linkou Chang Gung Memorial Hospital from 2011 to 2019. The treatment strategy for each patient was tailored according to multidisciplinary discussion, physician’s judgment, and shared decision-making with the patient. Patients received first-line PCT with fluoropyrimidine and platinum-based regimens. Targeted therapy was administered to patients with HER-2 positivity. Immunotherapy was not administered in the present study. The response of PCT was evaluated by physical examination, laboratory testing (including tumor markers), and imaging studies (such as computed tomography or sonography). Tumor response was defined using the Response Evaluation Criteria in Solid Tumors (RECIST) criteria [24]. Selected patients who met the criteria of complete response, partial response, or stable disease after PCT were referred to the multidisciplinary team (involving the medical oncologist, surgeon, gastroenterologist, radiologist, and radiation oncologist) for discussion of further treatment plans. Eligible and suitable patients underwent conversion surgery within 4 weeks following the last cycle of chemotherapy. Conversion surgery included standard radical gastrectomy and D2 lymph node dissection plus metastasectomy for the detectable lesion to achieve complete cytoreduction (R0 resection). For the metastatic lesion with complete response (undetectable) after PCT based on imaging findings, we did not perform extensive surgery (metastasectomy).

This retrospective study was approved by the Chang Gung Medical Foundation Institutional Review Board (No. 201801640B0C102). The requirement for informed consent was waived for this retrospective study, according to our institutional guidelines.

### 2.2. Data Collection and Definition

We retrospectively collected data on the clinical characteristics, operative details, and outcomes. Cancer stage was defined according to the 8th edition of the American Joint Committee on Cancer TNM staging system, which is based on evidence from endoscopy, intraoperative findings, computed tomography, and positron emission tomography [25]. OS was defined as the time from the initial diagnosis or the date of surgery to the date of patient’s death or last follow-up (31 May 2020). The median (range) follow-up time was 20.1 (6.9–97.5), 11.4 (2.1–60.4), and 5.5 (1.1–24.6) months for conversion surgery, in-front surgery plus PCT, and in-front surgery alone, respectively.

### 2.3. Statistical Analysis

The patients’ clinical records and parameters were compared using the Kruskal–Wallis test and post-hoc Dunn multiple comparisons test. For the clinical characteristics of patients who underwent conversion surgery, we used the chi-square test with Bonferroni adjusted post-hoc tests. Patients with in-hospital mortality were excluded from the survival analysis. OS rates were calculated using Kaplan–Meier curve analysis, and differences in survival time between the groups were assessed using the log-rank test.

All analyses were conducted using SPSS for Windows (version 20.0, IBM Corp., Armonk, NY, USA). Statistical significance was set at *p* < 0.05.

## 3. Results

The study enrolled 182 patients with mGC who underwent surgery. Among them, 25 patients (13.7%) underwent conversion surgery, 101 underwent in-front surgery (55.5%) plus PCT, and 56 (30.8%) underwent in-front surgery alone. Comparisons of the clinicopathological characteristics between the three groups are shown in Table 1. There were no significant differences in sex; type of gastrectomy; histology; metastatic pattern; and vascular, peritoneal, or lymphatic invasion and percentages of *Helicobacter pylori* infection among the groups. Patients who underwent in-front surgery alone were significantly older (*p* < 0.001) and had higher percentages of postoperative complications (*p* = 0.030) and in-hospital mortality (*p* = 0.006) compared to the other two groups. Kaplan–Meier survival curve analysis followed by the log-rank test revealed a significant difference in the OS rate (Figure 1; *p* < 0.0001). Patients who underwent conversion surgery had significantly higher OS rates than those who underwent in-front surgery plus PCT (*p* = 0.009) or in-front surgery alone (*p* < 0.0001). The 1- and 3-year survival rates were 87.1%, 53.1%, and 12.8%, respectively, and 31.4%, 16.4%, and 0%, respectively, for conversion surgery, in-front surgery plus PCT, and in-front surgery alone (median OS, 23.4 vs. 13.7 vs. 5.6 months).

Table 2 shows the clinicopathological characteristics of the patients in the conversion surgery group. The median age was 59.0 years, with 14 (56.0%) men and 11 (44.0%) women. Tumors were located in the stomach (96.0%) and esophagocardia junction (4.0%). Distant nodal metastases were identified in 12 (48.0%), peritoneal/omental metastases in 9 (36.0%), liver metastases in 5 (20.0%), and ovarian metastases in 3 (12.0%) patients. Downstaging (pathological stage I–III) was noted in 15 (60%) and non-downstaging in 10 (40%) patients. The median duration of chemotherapy before conversion surgery was 5.9 (range, 2.3–21.7) months. Table 3 shows the details of the therapeutic regimen for the conversion surgery group. All patients received fluoropyrimidine and platinum-based chemotherapies. Additional targeted therapy was administered to three patients with HER-2 positivity.

In the conversion surgery group, the median OS of 24 patients was 23.4 months after the initial diagnosis (95% confidence interval [CI], 17.9–28.9) and 14.2 months after the operation (95% CI, 7.8–20.6), respectively (Figure 2A,B). The 1-, 3-, and 5-year OS rates were 87.1%, 31.4%, and 31.4%, respectively, after the initial diagnosis and 59.5%, 33.5%, and 33.5%, respectively, after the operation. Patients with downstaging had a significantly better OS after the initial diagnosis than those without downstaging (Figure 3A; *p* = 0.016). The 1-, 2-, and 3-year OS rates were 100%, 54.2%, and 45.1%, respectively, and 66.7%, 27.8%, and 0%, respectively, in the downstaging vs. non-downstaging groups. Figure 3B shows that the median OS after the surgery was significantly longer in patients with downstaging than in patients without downstaging (19.4 vs. 5.8 months; *p* = 0.011).

We also analyzed the impact of metastasis site on survival in the conversion surgery group. Patients without distant node metastasis had better prognosis than those with distant node metastasis (Figure 4A; *p* = 0.021). In contrast, there were no significant differences in patient outcomes in terms of peritoneal/omental (Figure 4B; *p* = 0.418), liver (Figure 4C; *p* = 0.093), or ovarian metastasis (Figure 4D; *p* = 0.488).

## 4. Discussion

Despite recent developments in chemotherapy and even with the addition of targeted monoclonal antibody or immunotherapy to conventional chemotherapy, the prognosis of mGC is still unsatisfactory [3,4,5,6,7,8,26,27]. Our current study aimed to explore the surgical treatment strategy for mGC. In-front surgery usually prevents mGC-related complications, such as bleeding, obstruction, or perforation [10,12,28]. Studies have shown that in-front palliative resection can prolong survival in select patients [10,12,29]. However, chemotherapy may be interfered with, or postponed due to, surgical morbidities or surgery-related worsening of the patient’s general condition. A conversion surgery strategy can substantially prevent the immediate complications associated with in-front surgery and introduce properly timed therapy for patients with better performance. Our present results provide evidence that conversion surgery significantly prolonged the median OS compared with in-front surgery plus PCT or in-front surgery alone. Patients with downstaging had a better OS than those without downstaging.

Our data also showed that in-front surgery alone had significantly higher rates of mortality/complications and shorter OS time than conversion surgery or in-front surgery plus PCT, indicating that systemic treatment is the standard of care for mGC. Postoperative complications are independent risk factors for the early recurrence of GC after curative resection, which is related to the inflammatory cytokine cascade-induced dysfunction of immune cells, including cytotoxic T-lymphocytes, natural killer cells, and antigen-presenting cells, and a delay in the initiation of postoperative chemotherapy [30,31,32]. Postoperative complications and surgery-related immune suppression are major concerns in in-front surgery for mGC. Although there was no difference in mortality or complication rates between conversion surgery and in-front surgery plus PCT in our data, conversion surgery was associated with a significantly longer median OS than in-front surgery plus PCT. In addition, the median survival time between diagnosis and conversion surgery and after operation were 8.0 and 14.2 months, respectively, in the conversion surgery group. In the subgroup analysis, the median OS was as long as 19.4 months postoperatively in patients with downstaging; however, the median OS was only 5.8 months after surgery in non-downstaging patients. Based on our findings, we suggest that in-front surgery should be reserved for uncontrolled tumor-related complications in patients with mGC.

There are still several unmet needs regarding the treatment of advanced GC. Patients with mGC receive multiple-line chemotherapy, with or without a combination of targeted therapy or immunotherapy. However, the development of resistance or the cumulative adverse effects of therapeutic agents eventually prevent the continuation of treatment, and patients die from disease progression or tumor-associated complications [16,22,33]. Studies have also indicated that conversion surgery is feasible and prolongs patient survival for those who respond well to PCT with downstaging [22,33,34,35]. Similar to these previous findings, our current results demonstrated that patients with pathological stage I–III (downstaging) had a significantly higher median OS than those with stage IV (non-downstaging) after the initial diagnosis. The median OS after surgery was also significantly higher in the downstaging group.

It is important to appropriately select patients with mGC for conversion surgery. The criteria for the initial determination of “non-curativeness” or determination of resectability after PCT have not been established. Yoshida et al. suggested that marginal resectable tumors without peritoneal seeding are the best candidates for conversion therapy [35]. Furthermore, some factors have been proposed to predict prognosis and curative resectability in patients undergoing conversion surgery [21,22,34,36,37]. Studies have shown that histological tumor size and R0 resection are independent prognostic factors [21,36]. Yamaguchi et al. reported that 41.3 and 21.2 months of median survival were noted in patients that underwent R0 and R1/2 surgery, respectively [22]. Choe et al. showed significantly longer survival in patients with R0 resection than in those without R0 resection (did not reach median survival vs. 22.1 months) [34]. Furthermore, Kim et al. reported that curative conversion surgery was achieved in 10 of 43 patients (23.3%) with peritoneal seeding; they also noted that lymph node metastasis, an initial factor of non-curability, produced the best outcome, whereas peritoneal seeding was associated with a poor prognosis [37]. Choe et al. also noted that patients with a positive expression of NR2F1 (an orphan nuclear receptor that is a marker of cancer dormancy) in the initial biopsy specimen benefited from conversion surgery [34]. In our study, patients with distant node metastasis receiving conversion surgery experienced worse survival than those with non-distant node metastasis. However, these results should be interpreted with caution because the sample sizes of the abovementioned studies, including ours, were too small to draw solid conclusions. A randomized trial enrolling large-scale patients is needed to clarify the prognostic factors for selecting patients for conversion surgery with favorable outcomes.

A phase III randomized controlled study called the PRODIGY trial was designed to evaluate the survival benefits of neoadjuvant chemotherapy (NAC) in resectable GC compared with in-front surgery followed by chemotherapy, and it showed longer progression-free survival in the NAC group [38]. Interestingly, OS did not differ between the two groups. The inadequate density of postoperative treatments might, in part, explain the negative results. In addition, preoperative imaging studies assessing nodal status were suboptimal, leading to the overestimation of tumor stage, which might mainly benefit from surgery rather than chemotherapy. Nonetheless, this study provides insight into the effectiveness of chemotherapy in resectable advanced disease and does not detract from the effects of surgical treatment. In this regard, conversion surgery is reasonable for treating patients with downstaging after salvage therapy.

Our present results should be interpreted with caution. First, this retrospective study had an inherent selection bias that was unavoidable because of the small population. Second, patients who responded to salvage chemotherapy were not routinely sent for resection evaluation. We did not compare the outcomes of patients who underwent conversion surgery to those who were treated medically without surgery. Nonetheless, our study found that conversion surgery had a significantly higher 3-year survival rate than in-front surgery plus PCT and in-front surgery alone. Third, we did not perform diagnostic laparoscopy during the study period, and tissue proof of distant metastasis was not routinely obtained. Overstaging, which cannot be overlooked based on current modern diagnostic technology and which also exists in patients who do not undergo surgery, might partially explain the excellent outcomes of patients with downstaging in our current study. Fourth, we did not routinely test the expression of PD-L1, CDH1, and MSI status since immune checkpoint inhibitors were not reimbursed by our government health insurance during the period of this study.

In conclusion, our study suggests that conversion surgery provides survival benefits compared to in-front surgery plus PCT or in-front surgery alone in patients with mGC. Patients who underwent conversion surgery with downstaging had a better prognosis than those without downstaging. Non-curative-intent conversion surgery is not recommended because the survival gain is notably limited in non-downstaging patients. We recommend referring clinically mGC patients who respond to PCT to a multidisciplinary team for discussing further treatment plans (including conversion surgery) in order to improve survival outcomes. Further large-scale randomized trials are needed to validate the improved survival with conversion surgery by comparing mGC patients receiving conversion surgery to those treated only with non-surgical modalities.

## Figures and Tables

**Figure 1 jpm-12-00555-f001:**
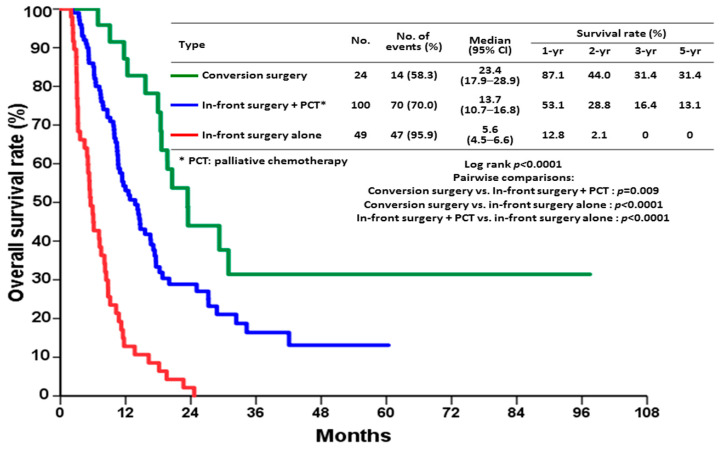
Kaplan–Meier survival curves in patients that underwent conversion surgery, in-front surgery plus palliative chemotherapy (PCT), and in-front surgery alone.

**Figure 2 jpm-12-00555-f002:**
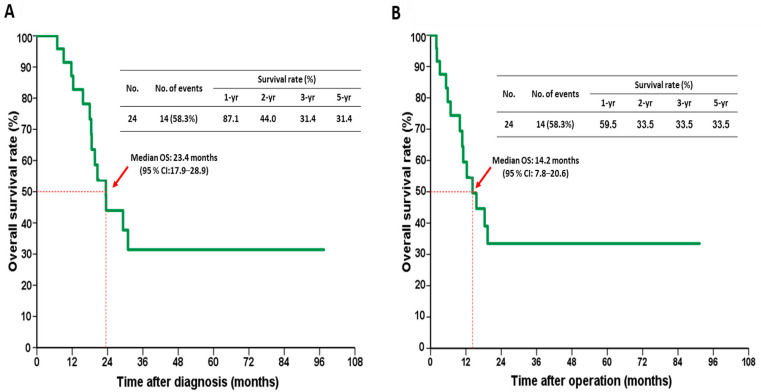
Kaplan–Meier survival curves after the (**A**) initial diagnosis and (**B**) operation in the conversion surgery group.

**Figure 3 jpm-12-00555-f003:**
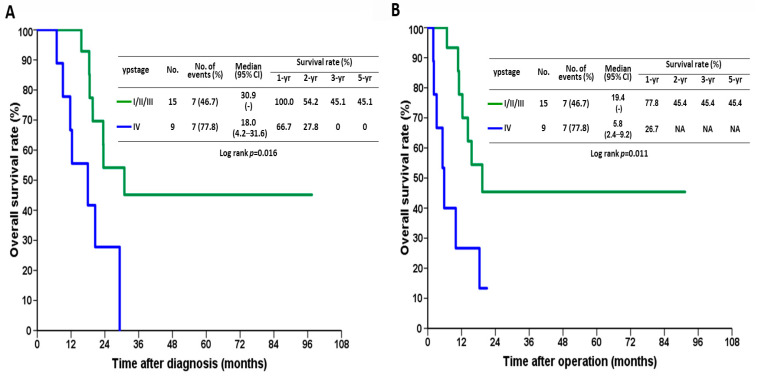
Kaplan–Meier survival curves after the (**A**) initial diagnosis and (**B**) operation in terms of yield pathological stage (ypstage).

**Figure 4 jpm-12-00555-f004:**
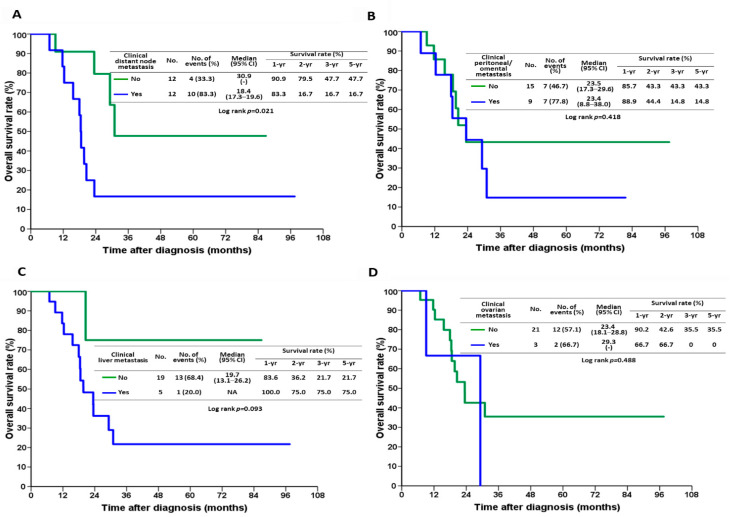
Kaplan–Meier survival curves after the initial diagnosis in the conversion surgery group in terms of (**A**) distant node metastasis, (**B**) peritoneal/omental metastasis, (**C**) liver metastasis, and (**D**) ovarian metastasis.

**Table 1 jpm-12-00555-t001:** Comparison of the clinicopathological parameters between patients that underwent conversion surgery, in-front surgery plus palliative chemotherapy (PCT), and in-front surgery alone.

Parameters	Conversion Surgery (*n* = 25)	In-Front Surgery Plus PCT (*n* = 101)	In-Front Surgery Alone (*n* = 56)	*p* Value
Age (years), median (IQR)	59 (15)	59 (16)	75 (18)	<0.0001
Sex				0.729
male	14 (56.0)	57 (56.4)	28 (50.0)	
female	11 (44.0)	44 (43.6)	28 (50.0)	
Gastrectomy				0.320
total	15 (60.0)	45 (44.6)	24 (42.92)	
partial	10 (40.0)	56 (55.4)	32 (57.1)	
Complication				0.030
yes	7 (28.0)	23 (22.8)	24 (42.9)	
no	18 (72.0)	78 (77.2)	32 (57.1)	
In-hospital mortality				0.006
yes	1 (4.0)	1 (1.0)	7 (12.5)	
no	24 (96.0)	100 (99.0)	49 (87.5)	
Histology				0.052
differentiated	10 (40.0)	18 (17.8)	15 (26.8)	
undifferentiated	15 (60.0)	83 (82.2)	41 (73.2)	
Metastatic patterns				0.071
hematogenous	16 (64.0)	40 (39.6)	22 (39.3)	
peritoneal seeding	9 (36.0)	61 (60.4)	34 (60.7)	
Vascular invasion				0.724
yes	12 (48.0)	50 (49.5)	24 (42.9)	
no	13 (52.0)	51 (50.0)	32 (57.1)	
Lymphatic invasion				0.797
yes	22 (88.0)	91 (90.1)	49 (87.5)	
no	3 (12.0)	10 (9.9)	7 (12.5)	
Perineural invasion				0.141
yes	19 (76.0)	84 (83.2)	39 (69.6)	
no	6 (24.0)	17 (16.8)	17 (30.4)	
*Helicobacter pylori* infection				0.878
yes	5 (20.0)	17 (16.8)	11 (19.6)	
no	20 (80.0)	84 (83.2)	45 (80.4)	

IQR: interquartile range.

**Table 2 jpm-12-00555-t002:** Clinicopathological characteristics of conversion surgery patients.

Parameters	No. of Patients	%
Sex		
male	14	56.0
female	11	44.0
Age (years), median (IQR)	59 (15)	
Tumor location		
esophagocardia junction	1	4.0
stomach	24	96.0
Site of distant metastasis		
distant node (paraaortic area, retroperitoneum, left supraclavicular area)	12	48.0
peritoneum/omentum	9	36.0
liver	5	20.0
ovary	3	12.0
Yield pathological stage		
I	3	12.0
II	3	12.0
III	9	36.0
IV	10	40.0
Duration of chemotherapy (months), median (range)	5.9 (2.3–21.7)	

IQR: interquartile range.

**Table 3 jpm-12-00555-t003:** Chemotherapy/targeted therapy regimens in patients that underwent conversion surgery.

Chemotherapy/Targeted Regimen	No. of Patients	%
XELOX (C2–13)	11	11.0
CCRT (XELOX; C12) followed by capecitabine (C6)	1	4.0
XELOX (C12) followed by capecitabine (C6, C8)	2	8.0
m-XELOX (C12)	2	8.0
m-XELOX (C12) followed by capecitabine (C5)	1	4.0
XELOX (C8) followed by XELOX/trastuzumab (C4)	1	4.0
XELOX/trastuzumab (C6) followed by capecitabine (C3)	1	4.0
Oxaliplatin/5-FU (C4, C9)	2	8.0
Oxaliplatin/5-FU/leucovorin (C3) followed by m-XELOX (C12)	1	4.0
PFL (C4)	1	4.0
Pertuzumab/trastuzumab/capecitabine (C13) followed by m-FOLFOX/ADI-PEG20 (C36)	1	4.0
FOLFOX (C18)	1	4.0

ADI-PEG20: pegylated arginine deiminase; C: cycle; CCRT: concurrent chemoradiotherapy; FOLFOX: fluorouracil/leucovorin/oxaliplatin; FP: fluorouracil/cisplatin; m-XELOX: modified XELOX; m-FOLFOX: modified FOLFOX; PFL: cisplatin/fluorouracil/leucovorin; XELOX: capecitabine/oxaliplatin; 5-FU: fluorouracil.

## Data Availability

Not applicable.

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
