# Peer review of "Outcomes of Conversion Surgery for Metastatic Gastric Cancer Compared with In-Front Surgery Plus Palliative Chemotherapy or In-Front Surgery Alone"

_jpm, 2022, doi:10.3390/jpm12040555_

Round 1

Reviewer 1 Report

Dear Authors,

Please improve your introduction

Please add more references to your research

Please add more conclusions

Kind regards

Author Response

March 27, 2022 

Journal of Personalized Medicine

RE: Manuscript ID:  jpm-1638991

Dear editor:

Thank you very much for your e-mail dated March 21, 2022, outlining the comments from the expert reviewers on our manuscript entitled “Outcomes of conversion surgery for metastatic gastric cancer compared with in-front surgery plus palliative chemotherapy or in-front surgery alone”. We appreciate the opportunity to revise our manuscript according to the referees’ suggestions.

I have listed the reviewers’ comments and provided responses to them accordingly. Changes suggested by the reviewers have been made in the revised manuscript and are highlighted in yellow.

I sincerely hope that the revised manuscript meets your team’s expectations and that it will now be suitable for publication in your esteemed journal.

Please let me know if there are any additional concerns relating to our manuscript.

Yours sincerely,

Jun-Te Hsu, MD

Professor, Department of General Surgery, Chang Gung Memorial Hospital at Linkou,

Chang Gung University College of Medicine

No. 5, Fushing Street, Kweishan Shiang District, Taoyuan City 333, Taiwan

Phone: 886-3-3281200, Ext: 3219

Fax: 886-3-3285818

Reviewer #1

Comments

Please improve your introduction

Please add more references to your research

Please add more conclusions

Responses: We thank the reviewer for the comments regarding our work. Accordingly, we have revised our manuscript and added content to the Introduction and Discussion sections. More references were also included in the revised manuscript. The changes have been highlighted in yellow. Please see the attachment.

Reviewer 2 Report

I think the authors did a nice job for this study and followed patients appropriately.

  1. It would be good for the authors to describe their approach to initiating surgery in their conversion group and when this was targeted. Was it after some specific time point or some algorithm for surgery. Moreover, was there a percentage decrease in tumor that led to surgery.
  2. The authors need to comment on their treatment of metastatic disease. Did they just perform surgery on the primary or did they perform more extensive surgery for liver or peritoneal disease.
  3. The authors need to discuss the MSI/PD-L1/CDH1/H pylori and other bio markers in the study group. I think this is really important as clearly different tumors respond differently based on their makeup

Author Response

March 27, 2022

Journal of Personalized Medicine

RE: Manuscript ID:  jpm-1638991

Dear editor:

Thank you very much for your e-mail dated March 21, 2022, outlining the comments from the expert reviewers on our manuscript entitled “Outcomes of conversion surgery for metastatic gastric cancer compared with in-front surgery plus palliative chemotherapy or in-front surgery alone”. We appreciate the opportunity to revise our manuscript according to the referees’ suggestions.

I have listed the reviewers’ comments and provided responses to them accordingly. Changes suggested by the reviewers have been made in the revised manuscript and are highlighted in yellow.

I sincerely hope that the revised manuscript meets your team’s expectations and that it will now be suitable for publication in your esteemed journal.

Please let me know if there are any additional concerns relating to our manuscript.

Yours sincerely,

Jun-Te Hsu, MD

Professor, Department of General Surgery, Chang Gung Memorial Hospital at Linkou,

Chang Gung University College of Medicine

No. 5, Fushing Street, Kweishan Shiang District, Taoyuan City 333, Taiwan

Phone: 886-3-3281200, Ext: 3219

Fax: 886-3-3285818

Reviewer #2

Comment 1: It would be good for the authors to describe their approach to initiating surgery in their conversion group and when this was targeted. Was it after some specific time point or some algorithm for surgery. Moreover, was there a percentage decrease in tumor that led to surgery.

Response: We thank the reviewer for the invaluable comments. We have added more detailed information regarding how to initiate conversion surgery and how to evaluate the therapeutic response in the Methods section. The response of PCT was evaluated by physical examination, laboratory testing (including tumor markers) and imaging studies, such as computed tomography or sonography. Tumor response was defined using the Response Evaluation Criteria in Solid Tumors (RECIST) criteria [24]. Selected patients who met the criteria of complete response, partial response, or stable disease after PCT were referred to the multidisciplinary team (involving the medical oncologist, surgeon, gastroenterologist, radiologist, and radiation oncologist) for discussing further treatment plans. Eligible and suitable patients underwent conversion surgery within 4 weeks following the last cycle of chemotherapy. Downstaging (pathological stage I–III) was noted in 15 (60%) and non-downstaging in 10 (40%) patients (Table 2).   

Comment 2: The authors need to comment on their treatment of metastatic disease. Did they just perform surgery on the primary or did they perform more extensive surgery for liver or peritoneal disease.

Response: Thank you for pointing this out. We have clarified our treatment strategy for conversion surgery in the Methods section. Conversion surgery included standard radical gastrectomy and D2 lymph node dissection plus metastasectomy for detectable lesion to achieve complete cytoreduction (R0 resection). For the metastatic lesion with complete response (undetectable) after PCT based on imaging findings, we did not perform extensive surgery (metastasectomy). Only one patient underwent oophorectomy.

Comment 3: The authors need to discuss the MSI/PD-L1/CDH1/H pylori and other bio- markers in the study group. I think this is really important as clearly different tumors respond differently based on their makeup

Response: Thank you for your comment. We completely agree with this that expression of PD-L1/CDH1, Helicobacter pylori infection, or MSI status may impact therapeutic effects or use of immunotherapy in gastric cancer. We added the results of Helicobacter pylori infection testing in Table 1 and the following sentences in the Results section: There was no difference in the percentages of Helicobacter pylori infection among groups. We did not routinely test the expression of PD-L1/CDH1 and MSI status since immune checkpoint inhibitors were not reimbursed by our government health insurance during the period of this study. This was included in the revised manuscript as a limitation of this study. In addition, immunotherapy was not administered in the present study. In the conversion surgery group, MSI status was determined in 11 patients with microsatellite stable (MSI-H: low). Microsatellite instability (MSI-H: high) was detected in two patients in the in-front surgery plus PCT group and in one patient in the in-front surgery alone group. PD-L1 expression was examined in two and six patients in the conversion surgery and in-front surgery plus PCT groups, respectively. We did not assess CDH1 expression.

                       Status

Conversion surgery

(n=25)

In-front surgery plus PCT

(n=101)

In-front surgery alone

(n=56)

MSI-H

 high

 low

 not available

0

11

14

2

34

65

1

14

41

PD-L1

 CPS <1%

 CPS >1%

 not available

1

1

13

1

5

95

0

0

56

Round 2

Reviewer 2 Report

Nice revisions.  No further issues